# Fourier-transform infrared spectroscopy for typing of vancomycin-resistant *Enterococcus faecium*: performance analysis and outbreak investigation

T. C. Scheier,[1] J. Franz,[1] M. Boumasmoud,[1,2] F. Andreoni,[1] B. Chakrakodi,[1] B. Duvnjak,[1] A. Egli,[3] W. Zingg,[1] A. Ramette,[4] A. Wolfensberger,[1] R. D. Kouyos,[1] S. D. Brugger[1]

**ABSTRACT**  Vancomycin-resistant Enterococci, mainly *Enterococcus faecium* (VREfm), are causing nosocomial infections and outbreaks. Bacterial typing methods are used to assist in outbreak investigations. Most of them, especially genotypic methods like multi-locus sequence typing (MLST), whole genome sequencing (WGS), or pulsed-field gel electrophoresis, are quite expensive and time-consuming. Fourier-transform infrared (FT-IR) spectroscopy assesses the biochemical composition of bacteria, such as carboxyl groups in polysaccharides. It is an affordable technique and has a faster turnaround time. Thus, the aim of this study was to evaluate FT-IR spectroscopy for VREfm outbreak investigations. Basic performance requirements like reproducibility and the effects of incubation time were assessed in distinct sample sets. After determining a FT-IR spectroscopy cut-off range, the clustering agreement between FT-IR and WGS within a retrospective (n: 92 isolates) and a prospective outbreak (n: 15 isolates) was investigated. For WGS an average nucleotide identity (ANI) cut-off score of 0.999 was used. Basic performance analysis showed reproducible results. Moreover, FT-IR spectroscopy readouts showed a high agreement with WGS-ANI analysis in clinical outbreak investigations (V-measure 0.772 for the retrospective and 1.000 for the prospective outbreak). FT-IR spectroscopy had a higher discriminatory power than MLST in the outbreak investigations. After determining cut-off values to achieve optimal resolution, FT-IR spectroscopy is a promising technique to assist in outbreak investigation as an affordable, easy-to-use tool with a turnaround time of less than one day.

**IMPORTANCE**  Vancomycin-resistant Enterococci, mainly *Enterococcus faecium* (VREfm), are a frequent cause of nosocomial outbreaks. Several bacterial typing methods are used to track transmissions and investigate outbreaks, whereby genome-based techniques are used as a gold standard. Current methods are either expensive, time-consuming, or both. Additionally, often, specifically trained staff needs to be available. This study provides insight into the use of Fourier-transform infrared (FT-IR) spectroscopy, an affordable, easy-to-use tool with a short turnaround time as a typing method for VREfm. By assessing clinical samples, this work demonstrates promising results for species discrimination and reproducibility. FT-IR spectrosopy shows a high level of agreement in the analysis of VREfm outbreaks in comparison with whole genome sequencing-based methods.

**KEYWORDS**  FT-IR spectroscopy, IR Biotyper, vancomycin-resistant enterococci, outbreak investigation

Vancomycin-resistant Enterococci (VRE) are a frequent cause of healthcare-associated infections, like urinary tract infections, surgical site infections, or bloodstream

Address correspondence to S. D. Brugger, silvio.brugger@usz.ch.

The authors declare no conflict of interest.

See the funding table on p. 12.

infections (1–3). In the last few years, there has been an increasing population-weighted mean percentage of vancomycin-resistant *Enterococcus faecium* (VREfm) isolates (2016: 11.6%; 2020: 16.8%) in the European Union and European Economic Area (4). Due to their resistance to vancomycin, they are associated with high mortality and increased health-care costs (1, 2, 5).

Since the main transmission route for VRE may be via direct or indirect contact, places crowded with individuals at risk for infections and colonization, such as healthcare institutions, are predisposed for outbreaks (6). This is also reflected by multiple hospital outbreaks worldwide (7–9).

The detection of outbreaks and tracking of transmissions are crucial for outbreak management and termination. For this reason, healthcare societies recommend that VRE isolates should be typed in certain patient populations as soon as clusters occur (10). Several typing methods exist, including whole genome sequencing (WGS), multi-locus sequence typing (MLST), and pulsed-field gel electrophoresis (PFGE). WGS is suggested as the preferred typing method for VRE, followed by PFGE and repetitive element sequence-based PCR or MLST. The latter are especially useful in long-lasting outbreaks (11). WGS shows high discriminatory power. Its use in daily routine is limited, because of high costs and a long turnaround time (12, 13). In contrast to genotypic methods, Fourier-transform infrared (FT-IR) spectroscopy offers a different, rapid, and low-cost approach by assessing the overall biochemical composition of cells throughout the whole infrared spectrum (4,000–400 $cm^{-1}$) (12, 14). For its use in cluster detection of bacteria, different wavenumber ranges can be assessed. The range of 1,200–900 $cm^{-1}$ seems to be promising as demonstrated, for example, for *Klebsiella pneumonia* (15). Standardized protocols for sample preparation are necessary since the amount of carbohydrates expressed can alter FT-IR spectra. Up to now, the interpretation of these data has also been restricted, as cluster-defining cut-offs have varied widely for different pathogens (12). Interpretation is also influenced by the variety of possible approaches that are employed as exploration algorithms (e.g., Euclidean distance) (15). imilar issues with the interpretation of data also exist for genome-based methods (11).

The aim of this study was therefore to assess the discriminatory power of FT-IR spectroscopy to determine bacterial clusters in VRE outbreaks compared with those of gold-standard methods. Reproducibility and diagnostic accuracy were investigated, including optimization of a reference cut-off value, which should facilitate the use of this method in daily routines. Specifically, we aimed to assess the performance of FT-IR spectroscopy compared with WGS in a retrospective and prospective outbreak investigation.

## MATERIALS AND METHODS

### Sample collection

Distinct sets of vancomycin-resistant Enterococci isolates, collected between 2011 and 2022 during routine analysis by the Department of Infectious Diseases and Hospital Epidemiology, University Hospital Zurich, were used for analysis. Frozen (−80℃) samples were streaked on Columbia Agar + 5% sheep blood (COS-plates) (bioMérieux SA, Marcy-l'Etoile, France) and incubated overnight at 36 ± 2℃. Susceptibility testing for all isolates was performed during routine clinical processing.

Species identification by matrix-assisted laser desorption ionization–time of flight (Bruker Daltonics, Bremen, Germany) was performed on all samples before FT-IR spectroscopy.

### Sample preparation

Three to five macroscopically identical colonies were subcultured on COS-plates for 18 ± 2 hours at 36 ± 2℃, unless stated otherwise. Following the proposed manufacturer's instructions, an overloaded 1-µL loop of bacterial colonies was suspended in a 50-µL

ethanol solution (70%, vol/vol), homogenized by vortexing, and mixed with 50 µL deionized water. Fifteen microliters of the homogenized suspension was placed on each of four spots (each spot consists of one technical replicate) on a 96-spot silicon plate (Bruker Daltonics, Bremen, Germany) and dried at 36 ± 2°C.

For sample set 6 (prospective outbreak), isolates were subcultured directly after identification during routine processing of the specimens.

Technical replicates were defined as samples obtained from one subculture of one isolate, while biological replicates were defined as samples obtained from different subcultures of the same isolate.

## Infrared measurements

An IR Biotyper (Bruker Daltonics) was used for all measurements. In each run, Bruker Infrared Test Standards 1 and 2 (ITRS 1/ITRS 2) were included to ensure the quality of the measurements.

Spectra in the wavenumber range for carbohydrates (1,300–00 cm$^{-1}$) were acquired, visualized, and processed by the OPUS v.8.2.28 software (Bruker Optik GmbH) and the IR Biotyper version 4.0.3.7334 (Bruker Daltonics). Only isolates with three or more replicates passing the quality control of the IR Biotyper were assessed. The Euclidean distance and UPGMA clustering method were used for data analysis.

For sample sets 5 and 6, isolates were assessed twice, once with the active and once with the inactive dimension reduction method (PCA 0.95/20) provided by the IR Biotyper.

Label coherence was used to assess the distribution of replicates of the same isolate within different clusters. By defining the isolate as the label, the IR Biotyper can display the clustering according to the label:

- Green: label is just present in one cluster.
- Yellow: label is present in two clusters.
- Orange: label is present in more than two clusters.

## Whole genome sequencing and comparative genomics

Whole genome sequencing was performed at the Institute of Medical Microbiology, University of Zurich, Switzerland, as previously described (16). In brief, after extraction of DNA, libraries were prepared using a MiSeq Reagent Kit (Illumina Inc., San Diego, CA) and sequenced on a MiSeq instrument (Illumina Inc.). Comparative genomic analyses were performed as previously described (17). Briefly, *de novo* assemblies were constructed with SPAdes v.3.10, and MLST was determined *in silico* using mlst v.2.7.6 (18, 19). This publication made use of the PubMLST website (https://pubmlst.org/) developed by Keith Jolley and sited at the University of Oxford (20). The development of that website was funded by the Wellcome Trust.

Average nucleotide identity (ANI) values were computed for each pair of genomes. Specifically, pairwise ANI values across whole genome sequences were calculated with pyani using the script average_nucletotide_identity.py with the command ANIb, which implements the method described by Goris and colleagues (21, 22).

## Comparison analysis of FT-IR spectroscopy and WGS

We defined FT-IR clusters based on hierarchical clustering using the Euclidean distance of the FT-IR spectra. FT-IR clusters were derived from the thereby obtained dendrograms by using different cut-offs. To assess the method's performance for outbreak investigation, we inferred transmission clusters based on comparative genomics. Two isolates were assumed to belong to the same transmission cluster if the ANI of their genomes exceeded a given threshold ANI $a_T$. ANI cut-off values range from 0 (non-identical) to 1 (identical). For the main analysis, we used an $a_T$ of 0.999. Finally, the overlap between ANI-based and FT-IR-based clusters was quantified using V-measure and an adjusted Rand-index.

## Performance evaluation

Distinct sample sets were used to assess the performance of the FT-IR spectroscopy:

- *In vitro* performance: four sample sets to investigate the discriminatory power at the species level, the impact of incubation time, the influence of the number of replicates, and reproducibility.
- Outbreak investigations: two sample sets to investigate a retrospective outbreak and a prospective outbreak.

### *In vitro* performance

### *Evaluation of different predefined cut-offs*

The FT-IR uses cut-offs to define clusters. If the distance between technical replicates is lower than the cut-off value, these replicates build up a cluster. Using a higher cut-off value results in reduced resolution.

The cut-off value was manually set to different values (0.11, 0.14, and 0.17) to approach the most reliable value for sample sets 1–4. The most reliable cut-off was defined as follows: sample set 1, best discrimination between VREfm and other Enterococci, and sample sets 2–4, best matching of the technical replicates of the same isolates, evaluated by label coherence. Label coherence was achieved if all replicates of the same isolates gathered within one cluster, independently of the mode or method of acquisition.

### *Sample set 1: species-level discrimination*

To evaluate FT-IR spectroscopy's performance at distinguishing isolates at the species level, five VREfm and 11 non-VREfm (seven *Enterococcus faecalis* and four *Enterococcus gallinarum*) were examined.

### *Sample set 2: incubation time*

Thirty-three isolates (31 VREfm and two *E. gallinarum*) were selected for evaluation of the impact of different incubation times on spectral clustering. Each strain was simultaneously streaked out on two plates. One plate was incubated for 18 ± 2 hours, the other for 24 ± 2 hours at 36 ± 2℃.

### *Sample set 3: number of technical replicates*

In the next step, we selected 20 VREfm isolates and carried out FT-IR spectroscopy using either four or 12 technical replicates. The different groups (4 replicates and 12 replicates) were processed and analyzed at different dates. Furthermore, the generated clusters were compared with each other. Sample processing was identical for the two groups, with a threefold increase in the volume and amount of bacteria for the 12-replicate group.

If clusters were present at any cut-off, the number and identity of isolates for each cluster were assessed for both groups. Cluster similarity was calculated by comparing the ID of the isolates in both groups and counting the identical ones.

### *Sample set 4: reproducibility*

To assess reproducibility, 11 VREfm isolates were analyzed three times within a 2-week period.

## Outbreak investigation

### Sample set 5: retrospective outbreak investigation

Ninety-two isolates collected during an outbreak from December 2019 to April 2020 were retrospectively used to compare FT-IR spectroscopy results with MLST and WGS results.

### Sample set 6: prospective outbreak investigation

A prospective, real-time outbreak investigation was performed using the IR Biotyper during the study. The results of the cluster detection and interpretation of transmission were compared with the results of MLST and WGS analysis, as soon as available.

## RESULTS

### Sample set 1: *species-level discrimination*

To test whether identification of Enterococci at the species level was possible using FT-IR spectroscopy, a pilot set of five VREfm isolates was compared with those of seven *E. faecalis* and four *E. gallinarum* (Fig. 1).

   By applying any of the predefined cut-offs, all tested VREfm formed one single cluster (Fig. 1, black box). To achieve clustering of *E. faecalis* and *E. gallinarum*, a higher threshold was needed. By choosing 0.317 as a cut-off, all isolates, except one *E. faecalis* isolate,

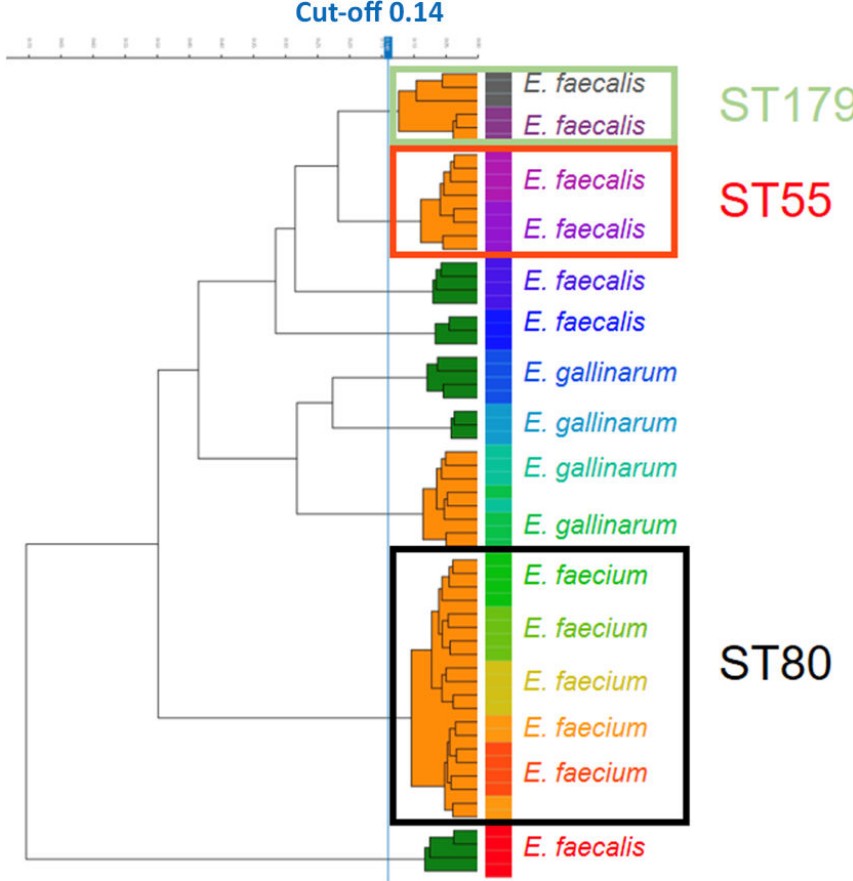

**FIG 1** Sample set 1: species-level discrimination. Dendrogram acquired by FT-IR spectroscopy of 16 isolates from three different Enterococci species. Isolates are labeled in different colors. Clusters are either displayed in orange (consisting of multiple isolates) or green (consisting of all technical replicates of one isolate) according to the applied cut-off (blue line; value: 0.14). Black box, cluster of *E. faecium*; green box, *E. faecalis* ST179 cluster; red box, *E. faecalis* ST55 cluster.

formed a distinct cluster according to their species. This supported the use of FT-IR spectroscopy to discriminate VREfm from other Enterococci species.

All VREfm were MLST80. The *E. faecalis* isolates were divided into five different MLST groups [ST179 ($n = 2$), ST55 ($n = 2$), ST774 ($n = 1$), ST40 ($n = 1$), and ST23 ($n = 1$)], whereby both isolates of ST179 and ST55 clustered together accordingly (Fig. 1, green and red boxes).

## Sample set 2: *incubation time*

To evaluate the impact of two different incubation times (18 and 24 hours) on isolate clustering, a total of 33 (31 VREfm and two *E. gallinarum*) isolates were assessed.

For both incubation times, clustering of the isolates depended on the applied predefined cut-offs (0.11, 0.14, and 0.17), with an increased congruence of the replicates at a higher cut-off. Fourteen (42%), 29 (88%), and 31 (94%) out of the 33 isolates included in the analysis had all their technical replicates from both biological replicates cluster together at a cut-off of 0.11, 0.14, and 0.17, respectively. At cut-off 0.17, the two isolates with different clustering at 18 and 24 hours of incubation time showed a clear separation of the biological replicates regarding the incubation time (Fig. 2, Isolates A and B). The two *E. gallinarum* isolates formed a separate cluster, which included all their technical replicates at cut-off values of 0.14 and 0.17.

## Sample set 3: *number of technical replicates*

By comparing 4 and 12 replicates for each isolate, conducted in different runs, several clusters were detected (Fig. S1). At a cut-off of 0.11, two clusters were detected for both replicate groups (Clusters A-0.11–4R/12R: six isolates; Clusters B-0.14–4R/12R: three isolates) with a cluster similarity of 83% (5/6 isolates) for Cluster A-0.11–4R and A-0.11–12R and 100% (3/3 isolates) for Cluster B-0.11–4R and B-0.11–12R. By increasing the cut-off to 0.14, the 4-replicate group just showed one cluster (A-0.14–4R: eleven isolates) and the 12-replicate group showed still two clusters (A-0.14–12R: ten isolates; B-0.14–12R: three isolates). A-0.14–4R included all 10 isolates of A-0.14–12R, but also an additional isolate, which leads to a cluster similarity of 91% (10/11). The three isolates (Isolates L, M, and N) of B-014–12R are composed of two different MLST (ST80 and ST117). At cut-off 0.17, one cluster exists for both replicate groups (A-0.17–4R: 12 isolates; A-0.17–12R: 14 isolates). A-0.17–12R includes all twelve isolates of A-0.17–4R and, in addition, two more (similarity 86%, 12/14). All isolates in A-0.17–4R are ST80; A-0.17–12R consists of two different MLST (80 and 117). All other isolates were singletons.

## Sample set 4: *reproducibility*

To test reproducibility over time, 11 VREfm isolates were run every seven days for a total of three times. Clustering was assessed by evaluating the percentage of isolates with all technical replicates within the same cluster. At a cut-off of 0.11, all replicates of four out of the 11 isolates clustered together. By increasing the cut-off to 0.14 or 0.17, all replicates of each isolate clustered together, independently of the timepoint of the run reflecting a 100% concordance regarding clustering (Fig. 3).

## Sample set 5: *retrospective outbreak investigation*

For the retrospective outbreak investigation, 92 isolates were analyzed, and WGS data were compared with clustering of FT-IR spectroscopy. Seven different MLST were detected by WGS: ST80 ($n = 72$), ST117 ($n = 10$), ST612 ($n = 5$), ST375 ($n = 2$), ST26 ($n = 1$), ST18 ($n = 1$), and ST 203 ($n = 1$).

By defining transmission clusters based on WGS data with an average nucleotide identity value cut-off above 0.999, five clusters were detected consisting of 64 (C1; ST80), 7 (C2; ST117), 5 (C3; ST612), 3 (C4; ST80), and 2 (C5; ST80) isolates, respectively. Eleven isolates were singletons. Using a cut-off of 0.165 showed the best concordance of the FT-IR spectroscopy data (without dimension reduction) to ANI data with an ANI cut-off value

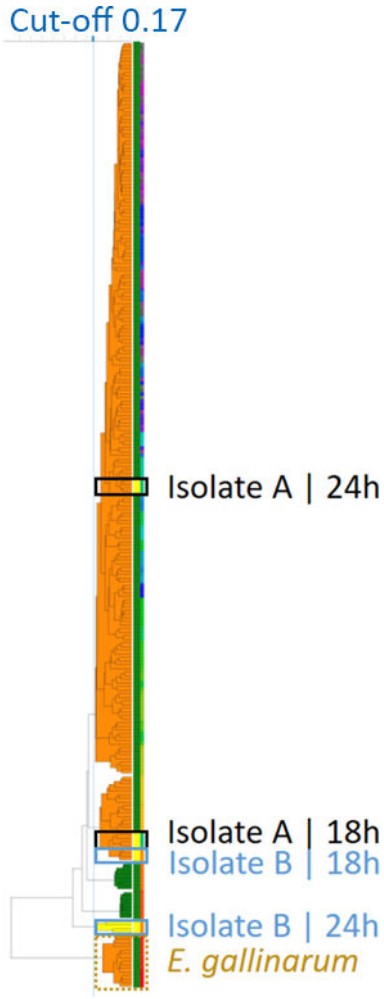

Cut-off 0.17

Isolate A | 24h

Isolate A | 18h
Isolate B | 18h

Isolate B | 24h
*E. gallinarum*

**FIG 2** Sample set 2: incubation time. Dendrogram acquired by FT-IR spectroscopy of 33 isolates to assess different incubation times at a cut-off value of 0.17 (blue line). All replicates of all isolates, except two isolates (A and B), belong to the same cluster. Black box, Isolate A; blue box, Isolate B; gold dashed line, *E. gallinarum*.

of 0.999 (V-measure: 0.772) (Fig. 4; see Fig. S2A and S3A for sensitivity analysis). Ten out of the 64 (C1) isolates were not correctly identified as being part of the ANI cluster using FT-IR spectroscopy. Out of these ten isolates, five isolates (Fig. 4: ID: 2) and two isolates (*F4:* ID: 9) formed their own cluster. One misclassified isolate of C1 (*F4*: ID: 14) formed a cluster with an isolate outside of C1. One isolate (*F4:* ID: 4) was misclassified as a singleton. Overall, 5/7, 4/5, and 2/3 isolates were correctly identified as belonging to C2, C3, and C4, respectively. C5 was correctly identified by FT-IR spectroscopy. One singleton isolate from the ANI analysis was misclassified as part of C1 by FT-IR spectroscopy. Two singletons (*F4:* ID: 5) were misclassified to be part of C3. A cluster generated by the FT-IR spectroscopy consisting of two isolates (*F4:* ID: 5) did not form a cluster according to the applied ANI cut-off. Not all technical replicates of two isolates (*F4:* ID: 0) belonged to the same cluster.

Increasing the FT-IR spectroscopy cut-off from 0.1 to 0.3 reduced the number of clusters and singletons from 43 to three (Fig. S4). Notably, a range of FT-IR spectroscopy cut-off between 0.135 and 0.245 produced very similar overlaps with the ANI clusters (V-measures > 0.700), such that these cut-offs can be considered as equally well performing (Fig. S5). Within this range, the choice of the cut-offs should be guided by the usual tradeoff between minimizing the probability of incorrectly assigning an isolate to a cluster while minimizing the chance of not assigning an isolate to a cluster.

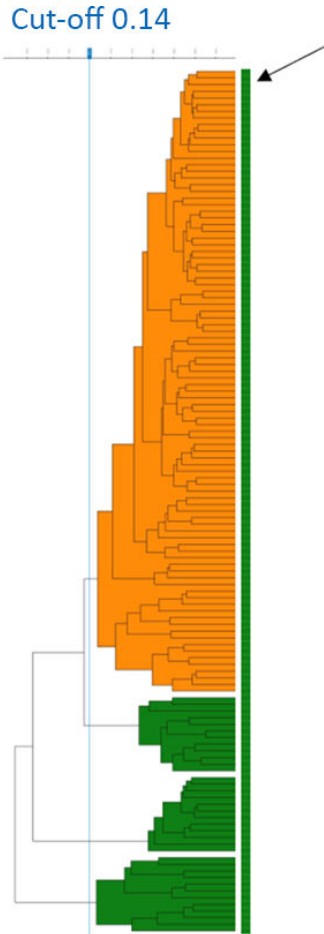

Cut-off 0.14

**FIG 3** Sample set 4: reproducibility. Dendrogram acquired by FT-IR spectroscopy of 11 isolates to assess reproducibility at different timepoints over a period of 2 weeks. Green bar (black arrow) represents the label coherence and indicates that all replicates of one isolate are within the same cluster. Cut-off: 0.14.

By applying the dimension reduction method (PCA 0.95/20) provided by the IR Biotyper, the optimized FT-IR spectroscopy cut-off value decreased to 0.135 to achieve a V-measure of 0.756 (Fig. S6A and S2B; sensitivity analysis).

### Sample set 6: *prospective outbreak investigation*

To assess the use of FT-IR spectroscopy in an outbreak situation within the hospital, we assessed 15 isolates sampled from 15 different patients (13 VREfm, 1 *E. faecalis,* and *1 E. gallinarum* isolates). According to WGS data, six different MLST clusters were present for *E. faecium*: ST203 ($n = 7$), ST80 ($n = 2$), ST375 ($n = 1$), ST721 ($n = 1$), ST117 ($n = 1$), and ST78 ($n = 1$). By using an ANI cut-off value of 0.999, seven isolates clustered together, while all others remained singletons (Fig. 5).

We compared the FT-IR spectroscopy cut-off values to the ANI cut-off values for cluster detection by assessing the adjusted rand index and the V-measure. Using a cut-off of 0.14 led to a 100% agreement of the FT-IR spectroscopy (without the dimension reduction method) with an ANI of 0.999 or higher, as presented in the overlapping network analysis (Fig. 5). For both measures, the optimal IR Biotyper cut-off was within the range of 0.14 to 0.175 (Fig. S3B and S7A for sensitivity analysis). When applying cut-offs outside of the range of 0.14 and 0.175, discordance could be detected, either by false identification of a cluster within this sample (cut-off 0.1) or by a missing discrimination outside of the cluster (cut-off 0.3) (Fig. S8 and S9).

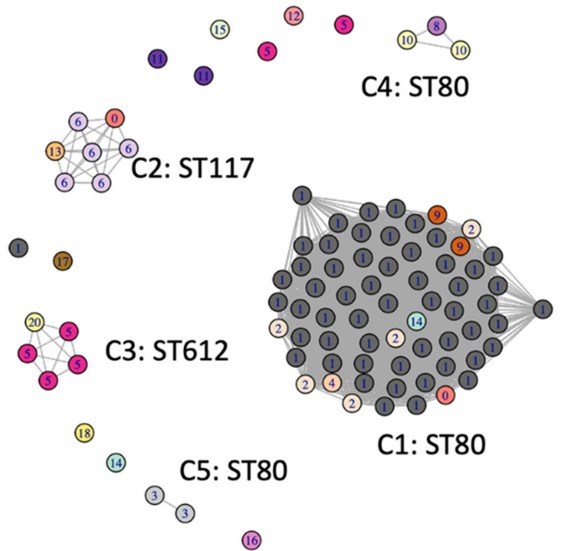

ANI cut-off value: 0.999
FT-IR spectroscopy cut-off: 0.165
V-measure: 0.772

FIG 4   Sample set 5: retrospective outbreak investigation. Network analysis: comparison of average nucleotide identity (ANI) networks with FT-IR spectroscopy clustering at cut-off 0.165. Each circle represents one isolate. Connected circles indicate an ANI cut-off value of 0.999 or higher. The numbers and colors of the circle represent the FT-IR spectroscopy cluster.

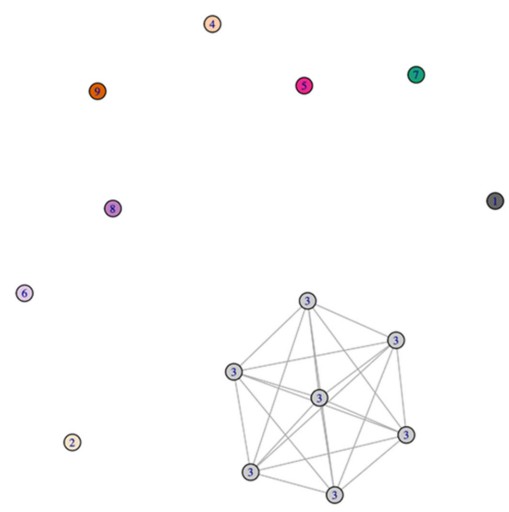

ANI cut-off value: 0.999
FT-IR spectroscopy cut-off: 0.140
V-measure: 1.000

FIG 5   Sample set 6: prospective outbreak investigation. Network analysis: comparison of average nucleotide identity (ANI) networks of prospective outbreak investigation with FT-IR spectroscopy at cut-off 0.140. Each circle represents one isolate. Connected circles indicate an ANI cut-off value of 0.999 or higher. The numbers and colors of the circle represent the FT-IR spectroscopy cluster.

By applying the dimension reduction method (PCA 0.95/20) provided by the IR Biotyper, the optimized FT-IR spectroscopy cut-off value decreased to 0.115 to achieve a V-measure of 1.000 (Fig. S6B, sensitivity analysis, Fig. S7B).

## DISCUSSION

Our study demonstrated that FT-IR spectroscopy showed good coherence (V-measure: 0.772 and 1.000) for vancomycin-resistant *Enterococcus faecium* outbreak investigations compared with genotypic methods.

Several studies have already assessed FT-IR spectroscopy as a general tool for outbreak investigations for different bacterial species, but only a few have evaluated basic performance, such as reproducibility (12, 23–30). To address the performance in the predefined sample sets, we used different cut-off values (0.11, 0.14, and 0.17), which were chosen according to the automatically generated cut-offs of the IR Biotyper, our growing experience in previous test runs and previously published reports (31).

By using a cut-off of 0.317, species discrimination was achieved for most of the isolates (sample set 1). The most reliable results in the *in vitro* performance groups were achieved with cut-off values of 0.14 and 0.17.

The turnaround time for the FT-IR spectroscopy is mainly depending on the incubation time. Other studies assessed various incubation times of up to 48 hours (15, 16, 23, 28–30, 32–36). We aimed to use an 18 ± 2-hour incubation time to report results within 24 hours. Nevertheless, we showed that prolonging the incubation time affects the analysis slightly, although at a cut-off of 0.17, 93.9% of the isolates clustered identically. Similar to our findings, Hu and colleagues showed that FT-IR spectroscopy analysis of *K. pneumoniae* is influenced by different incubation times (24 hours vs 48 hours) on distinct media (15).

Most other studies used three to five technical replicates for each isolate, but some also reported convenient results with only one technical replicate (12, 23, 26, 34, 37). Twelve technical replicates have shown to be superior to four replicates regarding concordance of FT-IR spectroscopy and WGS by using an automatically generated cut-off (30). We used predefined cut-offs and could not observe increased discriminatory power by using 12 technical replicates for each isolate. However, we observed the emergence of clusters, which included isolates belonging to different MLST types.

Outbreaks can often be long lasting, and therefore, the reproducibility of the method needs to be demonstrated. For *E. faecium*, already convenient data exist, showing that most isolates consistently grouped together in three consequent runs (29). We demonstrated similar results for VREfm at a cut-off of 0.14 and 0.17 over a period of 14 days.

In general, the FT-IR spectroscopy cut-off is specific for each bacterial species (12). In our hands, reproducibility and incubation time showed the best coherence when cut-off values of 0.14 and 0.17 were applied.

Cut-off values published for other bacterial species, such as *Klebsiella pneumonia*, *Streptococcus pneumoniae*, *Enterobacter cloacae*, and *Acinetobacter baumannii*, were in the range of around 0.15 to 0.50, which is higher than our findings (12, 15, 32). This underlines the fact that for every species, an optimal cut-off needs to be defined.

Multiple genotypic methods are used for outbreak investigations. Lytsy and colleagues showed that a core genome-based ANI cut-off threshold of 0.985 can be successfully used for VRE strain delineation and that its discriminatory power was superior to MLST and PFGE (38). In this study, an even higher cut-off value of 0.999 was applied as the risk of evolutionary changes in the isolates we tested was negligible as isolation dates were close in both outbreak investigations.

The overall performance of FT-IR spectroscopy, compared with that of ANI by V-measure, was 0.772 in the retrospective cluster and 1.000 in the prospective cluster. The difference could be explained by the different sample sizes (sample set 5: $n = 92$; sample set 6: $n = 15$) as well as the heterogeneity of the sample sets.

Most isolates in the retrospective outbreak set belonged to ST80, reflecting the dominant outbreak MLST in our hospital. Despite the genetic relatedness of the ST80

isolates included in this study, FT-IR spectroscopy allowed the segregation of isolates into several clusters revealing the presence of multiple lineages, as confirmed by WGS. This corroborates the superior discrimination power of FT-IR spectroscopy as compared with MLST-based classification of outbreaks. Nevertheless, as compared with those of WGS analysis, not all isolates were assigned to the correct clusters (V-measure: 0.772), possibly leading to misidentification of some isolates and resulting in the wrong interpretation of transmission chains.

In the prospective outbreak sample set, there was a 100% concordance of the FT-IR spectroscopy and the ANI networks when using a FT-IR spectroscopy cut-off of 0.14. MLST showed a lower discriminatory power by creating a second cluster of ST80 in addition to the ST203 one. In line with our findings, Teng and colleagues also showed a high congruence of cluster composition by comparing WGS with FT-IR spectroscopy for *E. faecium* but concluded that the discriminatory power is too low for use in routine surveillance (29).

While investigating the retrospective and prospective outbreaks, we could not find an improvement in the V-measure by using the dimension reduction approach provided by the IR Biotyper. An ongoing reevaluation of the cut-off in concordance with genetic information is needed to optimize the cut-off to match changes in the local epidemiology, such as the appearance of new isolates in the healthcare system. The different optimal range of cut-off values in our retrospective and prospective sample sets also highlight this issue (Fig. S3).

While many studies already used FT-IR spectroscopy for the rapid identification and typing of different bacterial species, we assessed predefined strains, a retrospective outbreak, and used data from clinical samples collected during routine epidemiological surveillance (30, 32, 35). Moreover, we used a wide range of sequence types and assessed the use in clinical routine as a real-time typing method during an ongoing outbreak.

Our study has multiple limitations: first, it took place in a single center and only one instrument was used. Additionally, we did not specifically assess the time needed to get familiar with the FT-IR without previous microbiological knowledge. However, a trained microbiologist will be able to learn this technique in no more than a few weeks. Second, the influence of growth media was not further investigated. We also did not investigate further the reason why some isolates clustered differently by FT-IR spectroscopy as compared with ANI.

In summary, FT-IR spectroscopy provides promising results for species discrimination and reproducibility at a turnaround time of less than one day for VREfm. Furthermore, we demonstrated high agreement with WGS-ANI even at a very high cut-off value of 0.999. In a prospective outbreak investigation, FT-IR spectroscopy showed better congruence with WGS-ANI than with MLST.

## ACKNOWLEDGMENTS

We thank Chun-Chi Chang, Srikanth Mairpady Shambat, and Peter W. Schreiber for their thoughtful inputs regarding the study.

This work was funded the University Hospital Zürich (grant number INOV00121 to T.C.S.) and Clinical Reasearch Priority Program (CRPP) BacVivo from the University of Zürich (to S.D.B.).

The funders of the study had no role in the study design, data collection, data analysis, data interpretation, or writing of the report.

## AUTHOR AFFILIATIONS

[1]Department of Infectious Diseases and Hospital Epidemiology, University Hospital Zürich, University of Zurich, Zurich, Switzerland
[2]Institute of Integrative Biology, ETH Zürich, Zurich, Switzerland
[3]Institute of Medical Microbiology, University of Zurich, Zurich, Switzerland
[4]Institute for Infectious Diseases, University of Bern, Bern, Switzerland

## AUTHOR ORCIDs

T. C. Scheier http://orcid.org/0000-0001-7805-1025
A. Ramette http://orcid.org/0000-0002-3437-4639
S. D. Brugger http://orcid.org/0000-0001-9492-9088

## FUNDING

| Funder | Grant(s) | Author(s) |
|---|---|---|
| University Hospital Zurich | INOV00121 | T. C. Scheier |
| UZH CRPP | BacVivo | S. D. Brugger |

## AUTHOR CONTRIBUTIONS

T. C. Scheier, Conceptualization, Data curation, Formal analysis, Funding acquisition, Investigation, Methodology, Project administration, Validation, Writing – original draft, Writing – review and editing | J. Franz, Data curation, Formal analysis, Investigation, Writing – review and editing | M. Boumasmoud, Formal analysis, Writing – review and editing, Methodology, Visualization | F. Andreoni, Conceptualization, Data curation, Investigation, Writing – review and editing | B. Chakrakodi, Data curation, Investigation | B. Duvnjak, Data curation, Investigation | A. Egli, Data curation, Methodology, Visualization | W. Zingg, Data curation, Methodology | A. Ramette, Conceptualization, Methodology, Writing – review and editing | A. Wolfensberger, Data curation, Writing – review and editing | R. D. Kouyos, Conceptualization, Data curation, Investigation, Methodology, Visualization, Writing – review and editing | S. D. Brugger, Conceptualization, Data curation, Formal analysis, Investigation, Methodology, Project administration, Supervision, Validation, Visualization, Writing – original draft, Writing – review and editing

## ADDITIONAL FILES

The following material is available online.

### Supplemental Material

**Figure S1 (Spectrum00984-23-s0001.tif).** Sample set 3: Number of technical replicates.
**Figure S2 (Spectrum00984-23-s0002.tif).** Sample set 5: Assessment of best agreement.
**Figure S3 (Spectrum00984-23-s0003.tif).** Sample set 5+6: Assessment of best agreement.
**Figure S4 (Spectrum00984-23-s0004.tif).** Sample set 5: Comparison of ANI networks to FT-IR.
**Figure S5 (Spectrum00984-23-s0005.tif).** Sample set 5: Comparison of average nucleotide identity (ANI) networks with FT-IR spectroscopy clustering at different cut-offs providing a V-measure of >0.700.
**Figure S6 (Spectrum00984-23-s0006.tif).** Sample set 5+6: Comparison of average nucleotide identity (ANI) networks with FT-IR spectroscopy clustering at optimal cut-offs.
**Figure S7 (Spectrum00984-23-s0007.tif).** Sample set 6: Assessment of best agreement.
**Figure S8 (Spectrum00984-23-s0008.tif).** Sample set 6: Comparison of average nucleotide identity (ANI) networks with FT-IR spectroscopy clustering at different cut-offs providing a V-measure of 1.000.
**Figure S9 (Spectrum00984-23-s0009.tif).** Sample set 6: Comparison of average nucleotide identity (ANI) networks with FT-IR spectroscopy clustering at different cut-offs.

### Open Peer Review

**PEER REVIEW HISTORY (review-history.pdf).** An accounting of the reviewer comments and feedback.

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
