## [Reviewer comments · Microbiology Spectrum]

Microbiology Spectrum

Fourier-transform infrared spectroscopy for typing of vancomycin-resistant *Enterococcus faecium* – performance analysis and outbreak investigation

Thomas Scheier, Jessica Franz, Mathilde Boumasmoud, Federica Andreoni, Bhavya Chakrakodi, Branko Duvnjak, Adrian Egli, Walter Zingg, Alban Ramette, Aline Wolfensberger, Roger Kouyos, and Silvio Brugger

Corresponding Author(s): Silvio Brugger, Universitat Zurich

Review Timeline:

Submission Date:	March 9, 2023
Editorial Decision:	May 18, 2023
Revision Received:	June 4, 2023
Accepted:	August 1, 2023

Editor: María Guembe

Reviewer(s): The reviewers have opted to remain anonymous.

Transaction Report:

DOI: <https://doi.org/10.1128/spectrum.00984-23>

May 18, 2023

Dr. Silvio D. Brugger
Universitat Zurich
Zurich
Switzerland

Re: Spectrum00984-23 (Fourier-transform infrared spectroscopy for typing of vancomycin-resistant *Enterococcus faecium* - performance analysis and outbreak investigation)

Dear Dr. Silvio D. Brugger:

Thank you for submitting your manuscript to Microbiology Spectrum. As you will see your paper is very close to acceptance. Please modify the manuscript along the lines I have recommended. As these revisions are quite minor, I expect that you should be able to turn in the revised paper in less than 30 days, if not sooner. If your manuscript was reviewed, you will find the reviewers' comments below.

When submitting the revised version of your paper, please provide (1) point-by-point responses to the issues raised by the reviewers as file type "Response to Reviewers," not in your cover letter, and (2) a PDF file that indicates the changes from the original submission (by highlighting or underlining the changes) as file type "Marked Up Manuscript - For Review Only". Please use this link to submit your revised manuscript. Detailed instructions on submitting your revised paper are below.

Link Not Available

Sincerely,

María Guembe

Reviewer comments:

Reviewer #1 (Comments for the Author):

Review Scheier et al - FT-IR spectroscopy for typing VREfm - May 2023

This work aimed at evaluating the use of FT-IR spectroscopy in typing strains from outbreaks caused by vancomycin-resistant *E. faecium* compared to other currently used methods as MLST, WGS and ANI. The authors conducted a very nice study, with several technical aspects taken into account as the repeatability (number of technical replicates), reproducibility (same samples analysed at different times), and the impact of incubation time. To evaluate the FT-IR spectroscopy approach, two sets of strains from outbreaks were studied: one set from a retrospective outbreak, and one set from a prospective outbreak.

The statistical analysis were good, and the authors are aware of the weaknesses of the study. Especially, they were not able to obtain congruent results for the retrospective outbreak using FT-IR spectroscopy compared with the typing obtained by WGS. This problem was not encountered for the prospective set, but as the authors claimed, this could be due to difference in the sample size.

Another point rightly presented by the authors is the fact that only one laboratory and one instrument are involved in this study. As the cut-off value seems to be central in this kind of approach, and as this cut-off is different from one species to another, it could be major to check that the cut-off values used in this study would be different or not in another lab. As this technique would be used routinely worldwide if suitable, this point seems to me to be a key point.

Especially, how other colleagues working on *E. faecium* could know if they rather should use a cut-off of 0.317 or rather of 0.14-

0.17? (I refer here to the end of the second paragraph of the Discussion, page 15).

Accordingly, I feel that the abstract is a little too optimistic, as if read alone, the reader could feel that the FT-IR spectroscopy can today be readily used instead of other currently used techniques. I am not sure that it is already so "easy-to-use" as getting to grips with this technology also involves working out the best cut-off to limit errors. In the last sentence of the abstract I would then propose "The use of FT-IR spectroscopy is a promising tool to assist in outbreak..." instead of "can be used".

I also feel that the idea explained in this sentence page 16 "This underlines the fact that for every species an optimal cut-off needs to be defined" should be indicated in a way within the abstract.

Other minor comments:

- Please number lines in your manuscript to facilitate the review
- Please put all figures and legends at the very end of the manuscript, not inside the Results text
- Introduction page 3, middle of page: "WGS is suggested as the..." (remove "the")
- Introduction page 3, end of page: " have vary" => "have varied"?
- Mat and Meth page 5, I do not think that the sentence "The development of that website was founded by..." is required here, as this is not part of this work.
- Mat and Meth page 7, usually the section "Role of the funding source" is located at the end of the manuscript, not in the Mat and Meth section.
- Figure 1 and others: it is not possible to read the cut-off line (too small)
- Figure 1, I wondered why one ST of *E. faecalis* clusters very distantly? Was it expected? (sometimes some ST are historically linked to the species but are definitely different, if so, this could be an opportunity to propose a new species?)
- Figure 4, it could be nice to show with strains are supposed to correspond to ST375, ST26 and ST203.
- Discussion page 15, I propose "by using 12 instead of four technical replicates for each isolate because..."

Reviewer #2 (Comments for the Author):

The ms by Scheier et al. describes the use of infrared spectroscopy to identify strains of *Enterobacter*, especially those coming from Vancomycin resistant outbreaks.

The data are of interest and the comparisons appropriate. My main complaint is that someone who is not versed in FT-IR will be unable to take advantage of the experiments due to extensive use of jargon throughout the ms. I strongly suggest changing that. For example, define what is the meaning of a cut-off and how affects measurements. The comparison with whole genome sequencing seems to hold well, but again, the jargon will not permit any reader to appreciate the usefulness of the technique.

In the discussion, it will be good to add how much preparation time is needed for someone to become adept to interpret the data coming from the FT-IR to have a sense of the usefulness of the technique. Authors claim that this is better and quicker than many, but do not explain in a convincing way why this is the case.

Reviewer #3 (Comments for the Author):

This is a manuscript that attempts to evaluate FT-IR spectroscopy for VREfm outbreak investigations and also compare to MLST and WGS by using an average nucleotide identity (ANI) cut-off score of 0.999. The topic of the manuscript is of interest to the Journal. The authors should give more detailed information in the "Infrared measurements" to let other researchers be able to be familiar with detection procedures and replicate. Although reproducibility and diagnostic accuracy have been studied, other influencing factors such as bacterial concentration and pre-treatment methods have not been investigated. Although this study has tested whether identification of enterococci at the species level were possible using FT-IR spectroscopy, can FT-IR spectroscopy distinguish vancomycin-sensitive *Enterococcus faecium* from VREfm? The similar study has been published (Azrad M, Matok LA, Leshem T, Peretz A. Comparison of FT-IR with whole-genome sequencing for identification of maternal-to-neonate transmission of antibiotic-resistant bacteria. *Microbiol Methods*. 2022 Nov;202:106603.), the innovation of this study is not enough.

There are some other details and inconsistencies in the manuscript which should also be addressed. Some are listed below.

1. "sample set 2-4" should be "sample sets 2-4"
2. The VREfm format is inconsistent and is italicized in some places

Preparing Revision Guidelines

To submit your modified manuscript, log onto the eJP submission site at <https://spectrum.msubmit.net/cgi-bin/main.plex>. Go to Author Tasks and click the appropriate manuscript title to begin the revision process. The information that you entered when you first submitted the paper will be displayed. Please update the information as necessary. Here are a few examples of required

updates that authors must address:

Please return the manuscript within 60 days; if you cannot complete the modification within this time period, please contact me. If you do not wish to modify the manuscript and prefer to submit it to another journal, please notify me of your decision immediately so that the manuscript may be formally withdrawn from consideration by Microbiology Spectrum.

This work aimed at evaluating the use of FT-IR spectroscopy in typing strains from outbreaks caused by vancomycin-resistant *E. faecium* compared to other currently used methods as MLST, WGS and ANI. The authors conducted a very nice study, with several technical aspects taken into account as the repeatability (number of technical replicates), reproductibility (same samples analysed at different times), and the impact of incubation time. To evaluate the FT-IR spectroscopy approach, two sets of strains from outbreaks were studied: one set from a retrospective outbreak, and one set from a prospective outbreak.

The statistical analysis were good, and the authors are aware of the weaknesses of the study. Especially, they were not able to obtain congruent results for the retrospective outbreak using FT-IR spectroscopy compared with the typing obtained by WGS. This problem was not encountered for the prospective set, but as the authors claimed, this could be due to difference in the sample size. Another point rightly presented by the authors is the fact that only one laboratory and one instrument are involved in this study. As the cut-off value seems to be central in this kind of approach, and as this cut-off is different from one species to another, it could be major to check that the cut-off values used in this study would be different or not in another lab. As this technique would be used routinely worldwide if suitable, this point seems to me to be a key point. Especially, how other colleagues working on *E. faecium* could know if they rather should use a cut-off of 0.317 or rather of 0.14-0.17? (I refer here to the end of the second paragraph of the Discussion, page 15).

Accordingly, I feel that the abstract is a little too optimistic, as if read alone, the reader could feel that the FT-IR spectroscopy can today be readily used instead of other currently used techniques. I am not sure that it is already so “easy-to-use” as getting to grips with this technology also involves working out the best cut-off to limit errors. In the last sentence of the abstract I would then propose “The use of FT-IR spectroscopy is a promising tool to assist in outbreak...” instead of “can be used”. I also feel that the idea explained in this sentence page 16 “This underlines the fact that for every species an optimal cut-off needs to be defined” should be indicated in a way within the abstract.

Other minor comments:

- Please number lines in your manuscript to facilitate the review
- Please put all figures and legends at the very end of the manuscript, not inside the Results text
- Introduction page 3, middle of page: “WGS is suggested as the...” (remove “the”)
- Introduction page 3, end of page: “ have vary” => “have varied”?
- Mat and Meth page 5, I do not think that the sentence “The development of that website was founded by...” is required here, as this is not part of this work.
- Mat and Meth page 7, usually the section “Role of the funding source” is located at the end of the manuscript, not in the Mat and Meth section.
- Figure 1 and others: it is not possible to read the cut-off line (too small)
- Figure 1, I wondered why one ST of *E. faecalis* clusters very distantly? Was it expected? (sometimes some ST are historically linked to the species but are definitely different, if so, this could be an opportunity to propose a new species?)
- Figure 4, it could be nice to show with strains are supposed to correspond to ST375, ST26 and ST203.

- Discussion page 15, I propose "by using 12 instead of four technical replicates for each isolate because... "

Manuscript Number: Spectrum00984-23

Title: Fourier-transform infrared spectroscopy for typing of vancomycin-resistant *Enterococcus faecium*- performance analysis and outbreak investigation

Comments to the Authors

This is a manuscript that attempts to evaluate FT-IR spectroscopy for VREfm outbreak investigations and also compare to MLST and WGS by using an average nucleotide identity (ANI) cut-off score of 0.999. The topic of the manuscript is of interest to the Journal. The authors should give more detailed information in the "Infrared measurements" to let other researchers be able to be familiar with detection procedures and replicate. Although reproducibility and diagnostic accuracy have been studied, other influencing factors such as bacterial concentration and pre-treatment methods have not been investigated. Although this study has tested whether identification of *enterococci* at the species level were possible using FT-IR spectroscopy, can FT-IR spectroscopy distinguish vancomycin-sensitive *Enterococcus faecium* from VREfm? The similar study has been published (Azrad M, Matok LA, Leshem T, Peretz A. Comparison of FT-IR with whole-genome sequencing for identification of maternal-to-neonate transmission of antibiotic-resistant bacteria. Microbiol Methods. 2022 Nov;202:106603.), the innovation of this study is not enough.

There are some other details and inconsistencies in the manuscript which should also be addressed. Some are listed below.

1. "sample set 2-4" should be "sample sets 2-4"
2. The VREfm format is inconsistent and is italicized in some places

Point-by-point reply: Fourier-transform infrared spectroscopy for typing of vancomycin-resistant *Enterococcus faecium* – performance analysis and outbreak investigation

Reviewer 1:

This work aimed at evaluating the use of FT-IR spectroscopy in typing strains from outbreaks caused by vancomycin-resistant *E. faecium* compared to other currently used methods as MLST, WGS and ANI. The authors conducted a very nice study, with several technical aspects taken into account as the repeatability (number of technical replicates), reproductibility (same samples analysed at different times), and the impact of incubation time. To evaluate the FT-IR spectroscopy approach, two sets of strains from outbreaks were studied: one set from a retrospective outbreak, and one set from a prospective outbreak.

The statistical analysis were good, and the authors are aware of the weaknesses of the study.

Especially, they were not able to obtain congruent results for the retrospective outbreak using FT-IR spectroscopy compared with the typing obtained by WGS. This problem was not encountered for the prospective set, but as the authors claimed, this could be due to difference in the sample size.

Another point rightly presented by the authors is the fact that only one laboratory and one instrument are involved in this study.

Reply: We thank the reviewer for the positive feedback about the methodology of our study. Also, we appreciate that the reviewer highlights the issue concerning the single center setting (one laboratory and one instrument) as well as the importance of the cut-off value. This is an important limitation, as pointed out in the discussion (line: 429 and 430).

As the cut-off value seems to be central in this kind of approach, and as this cut-off is different from one species to another, it could be major to check that the cut-off values used in this study would be different or not in another lab. As this technique would be used routinely worldwide if suitable, this point seems to me to be a key point. Especially, how other colleagues working on *E. faecium* could know if they rather should use a cut-off of 0.317 or rather of 0.14-0.17? (I refer here to the end of the second paragraph of the Discussion, page 15).

Reply: Depending on the local epidemiology, a different level of resolution might be needed. If the cluster of interest is composed by isolates with different MLST types or even different species, a lower resolution (higher cut-off for FT-IR), can be applied, compared to a cluster with only one MLST. According to the currently available literature and our study, we propose a stepwise approach to narrow down an applicable cut-off value. First, the literature should be screened for proposed cut-off values for the

investigated species. After applying these cut-off values to the isolates, an assessment of the FT-IR results, in comparison to another method (preferably high resolution such as WGS) should take place. Afterwards, the cut-off can be readjusted and individualized according to the needed resolution. Ongoing reassessments might be needed if the local epidemiology changes. This was also the approach we used for the study: At the beginning of our experiments, we applied different cut-off values (published and our experience) and evaluated them through the “in vitro performance” experiments. Afterwards we compared them to WGS, in a retro- and afterwards in a prospective setting. We hope that the manuscript offers some guidance of this approach, including set ups which can be used to approach a suitable cut-off, before it can be optimized even further as soon as WGS data might be available.

Accordingly, I feel that the abstract is a little too optimistic, as if read alone, the reader could feel that the FT-IR spectroscopy can today be readily used instead of other currently used techniques. I am not sure that it is already so “easy-to-use” as getting to grips with this technology also involves working out the best cut-off to limit errors. In the last sentence of the abstract I would then propose “The use of FT-IR spectroscopy is a promising tool to assist in outbreak... ” instead of “can be used”.

I also feel that the idea explained in this sentence page 16 “This underlines the fact that for every species an optimal cut-off needs to be defined” should be indicated in a way within the abstract.

Reply: We want to thank the reviewer for this input. The term “easy-to-use” was meant to indicate the simple workflow without the need of highly trained staff. We acknowledge that this can be misleading and changed the term as proposed and integrated the limitation of a species-specific cut-off value.

Previous version:

Line 57 and 58:

The use of FT-IR spectroscopy can be used to assist in outbreak investigation as an affordable, easy-to-use tool with a turn-around-time of less than one day

Revised version:

Line 61 – 63:

After determining cut-off values to achieve optimal resolution, FT-IR spectroscopy is a promising technique to assist in outbreak investigation as an affordable, easy-to-use tool with a turn-around-time of less than one day.

Other minor comments:

Reply: All the comments below are very helpful and we are grateful for this thorough review of the manuscript.

- Please number lines in your manuscript to facilitate the review

Reply: We implemented a numbering of the lines.

- Please put all figures and legends at the very end of the manuscript, not inside the Results text

Reply: All figures including their legends were moved to the end of the manuscript (track change version). We also revised the numbering of the figures because this was not correct. We apologize for the inconvenience.

Figure 4:

Previous version: Figure 5: Network analysis Sample set 5 (retrospective outbreak investigation). Comparison of average nucleotide identity (ANI) networks with FT-IR spectroscopy clustering at cut-off 0.165. Each circle represents one isolate. Connected circles indicate an ANI cut-off value of 0.999 or higher. Number and colors of the circle represent the FT-IR spectroscopy cluster.

Revised version: Figure 4: Network analysis Sample set 5 (retrospective outbreak investigation). Comparison of average nucleotide identity (ANI) networks with FT-IR spectroscopy clustering at cut-off 0.165. Each circle represents one isolate. Connected circles indicate an ANI cut-off value of 0.999 or higher. Number and colors of the circle represent the FT-IR spectroscopy cluster.

- Introduction page 3, middle of page: “WGS is suggested as the...” (remove “the”)

Reply: We corrected the spelling as proposed.

Line 98:

Previous version: WGS is suggested as *the* preferred...

Revised version: WGS is suggested as preferred...

- Introduction page 3, end of page: “ have vay” => “have varied”?

Reply: We corrected the spelling as suggested.

Line 110:

Previous version: defining cut-offs have *vary* widely for different

Revised version: defining cut-offs have *varied* widely for different

- Mat and Meth page 5, I do not think that the sentence “The development of that website was founded by...” is required here, as this is not part of this work.

Reply: We agree with the reviewer, but– if accepted by the reviewer – would not revise this citation because it is proposed like this. We respect the work of the cited authors and would therefore not change it.

<https://github.com/tseemann/mlst>:

“The mlst software incorporates components of the PubMLST database which must be cited in any publications that use mlst:

"This publication made use of the PubMLST website (<https://pubmlst.org/>) developed by Keith Jolley (Jolley & Maiden 2010, BMC Bioinformatics, 11:595) and sited at the University of Oxford. The development of that website was funded by the Wellcome Trust".

- Mat and Meth page 7, usually the section "Role of the funding source" is located at the end of the manuscript, not in the Mat and Meth section.

Reply: We deleted the paragraph in the Material and Methods section.

Previous version:

Role of the funding source

The funders of the study had no role in study design, data collection, data analysis, data interpretation, or writing of the report

Revised version: Lines 448 and 449 were added:

The funders of the study had no role in study design, data collection, data analysis, data interpretation, or writing of the report.

- Figure 1 and others: it is not possible to read the cut-off line (too small)

Reply: We are aware of this issue. In our opinion it would be not correct to add a wider line, since this would suggest wrong proportion or would cover some parts of the dendrogram and therefor indicate a different clustering of the isolates. We also think, pointing out the line by other forms (arrows, etc.) would also cover a part of the dendrogram and would distract the reader from the clustering assembly. Moreover, we tried to be in-line with the ASM policy to not change generated figures. In the initial submission we justified every change in an attached document.

Nevertheless, we approached this issue and included the cut-off value into the figure, on top of the line. We hope, this helps to highlight the cut-off line and indicates the cut-off value, without affecting the dendrogram. If the reviewer and the editor still think, it is needed to implement this, we are happy to provide the figures, but would suggest moving the changed figures into the supplementary because of the above mentioned considerations.

- Figure 1, I wondered why one ST of *E. faecalis* clusters very distantly? Was it expected?

(sometimes some ST are historically linked to the species but are definitely different, if so,

this could be an opportunity to propose a new species?)

Reply: We appreciate this comment and the input. Currently we can not answer this question with scientific evidence. We will therefore, besides other investigations, also incorporate the suggestion of the reviewer in our further experiments. We referred to this kind of issue in the limitation (line 433 – 435: We also did not investigate further the reason why some isolates clustered differently by FT-IR spectroscopy as compared to ANI). In the current manuscript we don't see a need to change either the figure or the text, because the evidence so far indicates that this is an *E. faecalis*.

- Figure 4, it could be nice to show with strains are supposed to correspond to ST375, ST26 and ST203.

Reply: We want to thank the reviewer for this thoughtful comment. Highlighting the isolates would lead to an overwhelmed figure, which could distract the reader and the focus of the figure is the comparison of cluster of WGS-ANI and FT-IR. All mentioned MLST (375, 26, 203) are isolates outside of the large cluster (C1). Both isolates with MLST 375 are not similar enough to achieve an ANI of 0.999 or higher and just one isolate has either MLST 26 or 203. Therefore, we think the additional benefit of highlighting some of the singletons outside of C1 does not outweigh the possible distraction of the reader. Also, here, we are providing the additional figure (if required by the editor and the reviewer) but would also suggest to move the newly added modified figure to the supplementary.

Reviewer 2:

The ms by Scheier et al. describes the use of infrared spectroscopy to identify strains of Enterobacter, especially those coming from Vancomycin resistant outbreaks.

The data are of interest and the comparisons appropriate. My main complaint is that someone who is not versed in FT-IR will be unable to take advantage of the experiments due to extensive use of jargon throughout the ms. I strongly suggest changing that.

Reply: First, we want to thank the reviewer for his comments about our manuscript. We are sure, that these inputs lead to an improvement of our study. To make the manuscript more comprehensible we added some explanation about the cut-off value (line 197-199) and information about assessment of distance and use of ANI values are already provided in the manuscript. We reduced the specific wording as much as possible. We hope to present the clinical key message (FT-IR is a promising tool to assist in outbreak management of VREfm) without providing a too technical manuscript or violating the word count.

Revised version: Lines 197 - 199 were added:

The FT-IR uses cut-offs to define clusters. If the distance between technical replicates is lower than the cut-off value, these replicates build up a cluster. Using a higher cut-off value results in a reduced resolution.

For example, define what is the meaning of a cut-off and how affects measurements. The comparison with whole genome sequencing seems to hold well, but again, the jargon will not permit any reader to appreciate the usefulness of the technique.

Reply: We think that by providing few additional explanations in the text, readers will understand the principles of our investigations. For further information the cited literature can be consulted. We added additional information, such as description of cut-off

In the discussion, it will be good to add how much preparation time is needed for someone to become adept to interpret the data coming from the FT-IR to have a sense of the usefulness of the technique. Authors claim that this is better and quicker than many, but do not explain in a convincing way why this is the case.

Reply: The FT-IR is an easy-to-use tool and we think a designated person can learn the whole process within a few weeks. This is based on our own experience with different technicians learning to perform FT-IR. However, in the present work, we didn't investigate this specifically. Therefore, we would prefer not to publish any statement about a specific time needed.

Previous version: Our study has multiple limitations: First, it took place in a single center and only one instrument was used.

Line 429 - 432 were added

Revised version: Our study has multiple limitations: First, it took place in a single center and only one instrument was used. Additionally, we did not specifically assess the time needed to get familiar with the FT-IR without previous microbiological knowledge. However, a trained microbiologist will be able to learn this technique in not more than a few weeks.

Reviewer 3

This is a manuscript that attempts to evaluate FT-IR spectroscopy for VREfm outbreak investigations and also compare to MLST and WGS by using an average nucleotide identity (ANI) cut-off score of 0.999. The topic of the manuscript is of interest to the Journal. The authors should give more detailed information in the "Infrared measurements" to let other researchers be able to be familiar with detection procedures and replicate. Although reproducibility and diagnostic accuracy have been studied, other influencing factors such as bacterial concentration and pre-treatment methods have not been investigated. Although this study has tested whether identification of enterococci at the species level were possible using FT-IR spectroscopy, can FT-IR spectroscopy distinguish vancomycin-sensitive *Enterococcus faecium* from VREfm? The similar study has been published (Azrad M, Matok LA, Leshem T, Peretz A. Comparison of FT-IR with whole-genome sequencing for identification of maternal-to-neonate transmission of antibiotic-resistant bacteria. *Microbiol Methods*. 2022 Nov;202:106603.), the innovation of this study is not enough. There are some other details and inconsistencies in the manuscript which should also be addressed. Some are listed below.

1. "sample set 2-4" should be "sample sets 2-4"
2. The VREfm format is inconsistent and is italicized in some places

Reply: We acknowledge the comments and critics of the reviewer. Despite this, we do not agree with all mentioned issues. First, we don't think we need to explain the FT-IR measurement further. A more detailed explanation would violate the word count and the clinical message of the paper. Additionally there multiple papers, including reviews and manual of the device, dealing with this theoretical issue, some of which we cited. Furthermore, the device we used (IR-Biotyper) is a commercially available diagnostic tool and we did not modify it or introduced a new technology. The manuscript should enable

the reader to understand the basics of the study (workflow, possible analysis, value in diagnostics), but also to gather further information by either the manufacturer or already published studies.

*We are aware that Azrad and colleagues assessed FT-IR for several species. In the paper, cited by the reviewer, **no assessment of VRE was done**. The results section presents data about *E. coli*, *K. pneumoniae* and MRSA (3.1., Typing of ESBL *E. coli*, 3.2. Typing of transmitted ESBL *Klebsiella pneumoniae*, 3.3. Typing of MRSA isolates). FT-IR, as shown by other groups, needs to be assessed for every species. Therefore, we think not that we can use any data proposed in the above-mentioned paper of Azrad.*

The main objective of this study was the performance of FT-IR for outbreak management of VRE. We did not plan to evaluate other probable diagnostic testing, such as AB testing, as discussed by the reviewer. We acknowledge that this would be an important work, and further studies should be conducted to address this question.

We incorporated the suggested changes into the revised version of the manuscript.

August 1, 2023

Dr. Silvio D. Brugger
Universitat Zurich
Zurich
Switzerland

Re: Spectrum00984-23R1 (Fourier-transform infrared spectroscopy for typing of vancomycin-resistant *Enterococcus faecium* - performance analysis and outbreak investigation)

Dear Dr. Silvio D. Brugger:

Your manuscript has been accepted, and I am forwarding it to the ASM Journals Department for publication. You will be notified when your proofs are ready to be viewed.

Sincerely,

María Guembe
Editor, Microbiology Spectrum
